



# Chemical composition, optical properties and radiative forcing efficiency of nascent particulate matter emitted by an aircraft turbofan burning conventional and alternative fuels

Miriam Elser[1,2], Benjamin T. Brem[1], Lukas Durdina[1], David Schönenberger[1], Frithjof Siegerist[3], Andrea Fischer[4], Jing Wang[1,2]

[1]Laboratory for Advanced Analytical Technologies, Empa, Dübendorf, 8600, Switzerland
[2]Institute of Environmental Engineering, ETH, Zürich, 8049, Switzerland
[3]SR Technics AG, Zurich-Airport, 8058, Switzerland
[4]Air pollution/Environmental Technology, Empa, Dübendorf, 8600, Switzerland

Correspondence to: Miriam Elser (miriam.elser@empa.ch)

**Abstract.** Aircraft engines are a unique source of carbonaceous aerosols in the upper troposphere. There, these particles can more efficiently interact with solar radiation than at ground. Due to the lack of measurement data, the radiative forcing from aircraft particulate emissions remains uncertain. To better estimate the global radiative effects of aircraft exhaust aerosol, its optical properties need to be comprehensively characterized. In this work we present the link between the chemical composition and the optical properties of the particulate matter (PM) measured at the engine exit plane of a CFM56-7B turbofan. The measurements covered a wide range of power settings (thrust), ranging from ground idle to take-off, using four different fuel blends of conventional Jet A-1 and Hydro-processed Ester and Fatty Acids (HEFA) biofuel. At the two measurement wavelengths (532 and 870 nm) and for all tested fuels, the absorption and scattering coefficients increased with thrust, as did the PM mass. The separation of elemental carbon (EC) and organic carbon (OC) revealed a significant mass fraction of OC (up to 90%) at low thrust levels, while EC mass dominated at medium and high thrust. The use of HEFA blends induced a significant decrease in the PM mass and the optical coefficients at all thrust levels. The HEFA effect was highest at low thrust levels, where the EC mass was reduced by up to 50-60%. The variability in the chemical composition of the particles was the main reason for the strong thrust dependency of the single scattering albedo (SSA), which followed the same trend as the OC fraction. Mass absorption coefficients (MAC) were determined from the correlations between aerosol light absorption and EC mass concentration. The obtained MAC values ($MAC_{532} = 7.5 \pm 0.3$ m$^2$ g$^{-1}$ and $MAC_{870} = 5.2 \pm 0.9$ m$^2$ g$^{-1}$) are in excellent agreement with previous literature values of absorption cross section for freshly generated soot. The Simple Forcing Efficiency (SFE) was used to evaluate the direct radiative effect of aircraft particulate emissions for various ground surfaces. The results indicate that aircraft PM emissions over highly reflective surfaces like snow or ice have a substantial warming effect. The use of the HEFA fuel blends decreased PM emissions, but no changes where observed in terms of EC/OC composition, optical properties and forcing per mass emitted.





## 1 Introduction

The rapid expansion of the aviation industry in the last decades and the continuous growth projected for the next 20 years (Leahy J., 2016), have motivated the study of aircraft engine emissions and their related effects on the environment and human health. Several field and modelling studies have investigated the degradation of air quality near airports and have

assessed the related effects on human health (e.g. Arunachalam et al. (2011), Barrett et al. (2013), Carslaw et al. (2006), Hsu et al. (2009), Lee et al. (2013) and Schürmman et al. (2007), among others). Aircraft engines are also a unique source of gases and particles in the upper troposphere and lower stratosphere, where they alter the atmospheric composition and contribute to climate change. In a study on the impacts of emissions from commercial aircraft flights on climate, Jacobson et al. (2013) reported that aircraft emissions were responsible for ~6 % of the Arctic surface global warming and ~1.3 % of

total surface global warming. The radiative forcing from aircraft emissions results from the direct release of radiatively active compounds (greenhouse gases and particulate matter (PM)), species that produce or destroy radiatively active substances (e.g. nitrogen oxides ($NO_x$) as an ozone precursor), and species that trigger the formation of condensation trails and cirrus clouds (Penner et al., 1999). Of special interest are the aerosol-light interactions by strongly absorbing black carbon (BC), which are known to cause positive radiative forcing (i.e. warming). Although aviation BC emissions are very

small relative to other anthropogenic sources like road transport, industry or biomass burning (Balkanski et al., 2010; Hendricks et al., 2004; Karagulian et al., 2017), their radiative effects can be enhanced when emitted at high altitude and over high surface albedo such as snow and ice surfaces or clouds. In fact, several model studies have shown that the direct radiation forcing (DRF) of BC strongly increases with altitude (e.g. Samset and Myhre, 2011; Zarzycki and Bond, 2010), and that globally, more than 40 % of the total DRF of BC is exerted at altitudes above 5 km (Samset et al., 2013). The

presence of clouds are a major contributor to the altitude dependency of the DRF of BC, but other factors such as surface albedo, water vapor concentrations and background aerosol distributions also contribute (Haywood and Shine, 1997; Samset and Myhre, 2011; Zarzycki and Bond, 2010). A detailed understanding of the optical properties of the carbonaceous particles emitted from aircraft exhaust is therefore essential to estimate the related climate effects.

Atmospheric PM scatters and absorbs solar radiation. Commonly reported optical parameters of PM include the

absorption and scattering coefficients ($b_{abs}$ and $b_{scat}$, respectively), and the single scattering albedo (SSA), defined as the ratio between light scattering and total extinction (absorption + scattering). The optical coefficients are often normalized to the particle' mass to provide the mass absorption and mass scattering cross sections (MAC and MSC, respectively), which are essential parameters in atmospheric radiative transfer models. The MAC, MSC and SSA are key optical properties for the assessment of the aerosols radiative effects, and strongly depend on the particle size, morphology, and chemical

composition. In the case of aircraft emissions, the characteristics of the PM emissions are influenced by the engine type, the thrust level (power) at which the engine is operated, and the fuel properties. The lack of experimental data on the optical properties of aircraft PM emissions has led to the extended use of generalized soot properties as an approximation to model aircraft radiative effects. This can lead to large discrepancies in the results, as the soot characteristics that determine the



optical properties significantly vary among different combustion sources and combustion conditions. For example, aircraft soot particles are characterized by a very high degree of crystallinity and low oxidative reactivity (Liati et al., 2016; Parent et al., 2016), which might affect light absorption properties. In their meticulous review work, Bond and Bergstrom (2007) suggested a consistent MAC for fresh light-absorbing carbon (MAC$_{\lambda=550nm}$ = 7.5 ± 1.2 m$^2$ g$^{-1}$), independent of the combustion

source or conditions. Higher MAC values were attributed to coating of the particles with negligibly-absorbing carbon. While in the size range of particles emitted from combustion processes the MAC stays nearly constant, the MSC has a strong dependency on the particle size (Hand and Malm, 2007), relative humidity (Khalizov et al., 2009), and coating with non-absorbing carbon (He et al., 2015). Levin et al. (2010) measured MSCs at 532 nm in the range 1.5 - 5.7 m$^2$ g$^{-1}$ for fresh biomass burning smoke from a variety of fuels. In addition, a previous study of biomass burning emissions from Reid et al.

(2005) reported a smaller range of MSC (3.2-4.2 m$^2$ g$^{-1}$) for fresh smoke, and larger MSC values for aged (coated) smoke (3.5-4.6 m$^2$ g$^{-1}$). The strong dependency of MAC and MSC (and therefore SSA) on the particles` size, morphology and chemical composition (which as indicated by our results strongly varies with thrust), makes the measurement of the optical properties of aircraft PM emissions a key step to decrease the uncertainty in the modelling of their radiative effects.

Concerns of the limited reserves of fossil fuels and the environmental impacts of their consumption have led to the

introduction of aviation biofuels. Compared to the standard Jet A-1 fuel, biofuels can have lower net CO$_2$ emissions. The use of biofuel blends in airliners is regulated by the ASTM D7566 (Standard specification for aviation turbine fuel containing synthesized hydrocarbons), which limits the maximum content of biofuel in the blend to 50% and sets restrictions to the blend aromatic content (minimum of 8%), lubricity, density, freezing point and viscosity (ASTM D7566-17a, 2017). One of the five ASTM certified blending components is biofuel from hydro-processed esters and fatty acids (HEFA), which can be

produced from any form of fat or oil (e.g. waste fats from food industry or vegetable oils and fatty acids from oil/fat refining processes). The main difference between HEFA fuel and conventional Jet A-1 fuel is the absence of sulfur and aromatic species, commonly present in Jet A-1 in the range of 10-1000 ppm of sulfur and around 18% of aromatic content (Hadaller and Johnson, 2006). Previous works have shown that reducing sulfur and aromatics in the fuel decreases the sulfate and BC emissions, respectively (Penner et al., 1999; Beyersdorf et al., 2014; Moore et al., 2015; Brem et al., 2015; Lobo et al.,

2016). Moreover, a recent study has shown that using biofuel blends to power aircraft engines reduces their particle emissions by 50 - 70% (Moore et al., 2017).

In this work we study the link between the chemical composition and the optical properties of the PM from aircraft exhaust for different engine loads and HEFA fuel blends. The measurements were performed using an in-service engine from the Boeing 737 Next Generation, which constitutes around 30% of all commercial airliners, and is therefore

representative of the current fleet. The chemical characterization of the exhaust was based on the separation of elemental and organic carbon (EC/OC analysis) from filter samples, while the optical properties were measured online at two different wavelengths. The resulting optical properties were then integrated in a simple two stream radiative model to estimate the direct forcing of particle emissions from aircraft turbines during cruise conditions. The Simple Forcing Efficiency (SFE,



Chylek and Wong, 1995) was evaluated over the entire solar spectrum for various surface albedos, including sea, grass, soil and snow.

## 2 Methods

### 2.1 Experimental set-up

The measurements were performed at the engine test cell of SR Technics at Zurich Airport (Switzerland) using an in-service commercial turbofan CFM56-7B burning four different blends of Jet A-1 and HEFA fuel (HEFA vol. percentage of 0, 5, 10, and 32%). A schematic of the experimental set-up is shown in Fig. 1. The exhaust was sampled at the engine exit plane using a single point sampling probe and then split into three sampling lines: the PM line for measurements of the particulate fraction, the GenTox line for the sampling of genotoxic compounds and the Annex 16 line for the measurements of the

gaseous emissions and smoke number. The PM line was diluted with dry synthetic air (dilution factor ∼ 1:10) to prevent water condensation and coagulation of the particles in the sampling line. The non-volatile PM (nvPM) measurement system is compliant with the new ICAO standard (ICAO, 2017). The instruments relevant for this work are shown in blue in Fig. 1. The chemical composition was determined from the analysis of filter samples with a Sunset OC-EC Aerosol Analyzer (Sunset Laboratory Thermal/Optical Carbon Analyzer, Model 4L). The optical properties were monitored online with a

Photo-Acoustic Extinctiometer (PAX, Droplet Measurement Technologies, $\lambda$=870 nm) and a Cavity Attenuated Phase Shift PM single scattering albedo monitor (CAPS $PM_{ssa}$, Aerodyne, $\lambda$=532 nm). The particle size distribution was measured using a Scanning Mobility Particle Sizer (SMPS, TSI, Model 3938). The BC mass concentration was measured using a Micro-Soot Sensor (MSS, AVL, Model 483). The $NO_2$ measurement (Eco Physics, CLD844 S hr) was used to perform an online correction of the interference in the optical measurements at 532 nm. The $CO_2$ analyzers (Thermo Fisher Scientific Model

410i in the PM line and Horiba PG-250 in the Annex 16 line) were used to calculate the dilution factors. Additional details of the measurement set-up are provided in the Supplementary Information (Sect. S1.1).

### 2.2 Filter samples for EC/OC analysis

Filter samples for EC/OC analysis were collected in the PM sampling line with a dual step stainless steel filter holder (URG, Series 2000-30FVT). Quartz fiber filters (Pall Tissuequartz, 2500QAT-UP) were used in both stages to collect the PM mass

(main filter) and to determine the positive sampling artifact (back-up filter) from gaseous OC adsorbing onto the filter surface (Kirchstetter et al., 2000, Subramanian et al., 2004). Prior to sampling, the filters were baked for at least 6 hours at 650 °C to remove possible contaminations of adsorbed carbon. The sampling flow was 5 l min$^{-1}$ and the duration was adjusted to provide optimal mass surface loadings for the EC/OC analyses (around 7 μg cm$^{-2}$). Stainless steel masks were deployed to reduce the sampling area of the filters and, as a result, increase the mass loading per sample area when needed.

Overall, 16 sets of filters were collected to cover the full range of thrust levels for the Jet A-1 fuel and the 32% vol. HEFA



blend. The larger filter masks (24 mm inner diameter) were used to sample at high thrust levels (100 - 65%), while the smaller masks (16 mm inner diameter) were required for the low thrust levels (50 - 7%).

The thermo-optical analysis for the quantification of EC and OC was performed using a Sunset OC-EC Aerosol Analyzer (Sunset Laboratory Thermal/Optical Carbon Analyzer, Model 4L). A detailed description of the method is reported in the Supplementary Information Sect. S1.2. For the analysis a modified NIOSH 5040 thermal protocol, summarized in Table S1 (Birch and Cary, 1996), with a transmittance optical correction for pyrolysis was used.

## 2.3 Measurement of the optical properties

The DMT PAX monitor simultaneously measures the aerosol optical absorption and scattering using a modulated diode laser ($\lambda$=870 nm). The light absorption is determined using the photo-acoustic technique. The modulated laser beam heats up the absorbing particles, which quickly transfer the heat to the surrounding air, generating a pressure wave that is measured with a sensitive microphone. The light scattering of the bulk aerosol is measured with a wide-angle integrating reciprocal nephelometer. Since there is relatively little absorption from gases and non-BC aerosol species at the 870 nm wavelength, the absorption measurement corresponds to the BC mass. Therefore, using an appropriate mass absorption cross section (MAC), the PAX absorption measurement can be used to determine BC mass ($BC_{PAX}$). Vice versa, comparing the absorption measurement with the EC mass from filter measurements, we can infer the $MAC_{870}$ for aircraft engine exhaust.

The CAPS $PM_{ssa}$ monitor provides simultaneous measurement of aerosols light extinction and scattering (Onasch et al., 2015). The extinction measurement is based on the cavity attenuated phase shift technique, which evaluates the phase shift of a LED light (532 nm) in a very long optical path (up to 2 km), created with very high reflectivity mirrors in the sampling cell (30 cm). In addition, the CAPS $PM_{ssa}$ includes an integrating sphere (integrated nephelometer) within the optical path for the measurement of particle scattering. A particle size dependent truncation correction is required to take into account the light lost at extreme forward and backward scattering angles due to the apertures of the optical beam.

Laboratory calibrations of both instruments were performed prior to the measurement campaign using size selected ammonium sulfate and nigrosine PM (see Sect. S1.3 in the Supplementary Information). In addition to the standard calibrations, corrections for the CAPS scattering signal outside the instrument linear range and for the interference from gaseous $NO_2$ at the measurement wavelength were also developed (Fig. S2 and S3). While the measured $NO_2$ interference in the CAPS extinction is fairly consistent with previously reported values, we also found an unexpected non-linear interference in the CAPS scattering signal. This was initially attributed to a possible light leak in the instrument, but the scattering interference persisted after the light sealing of the instrument was renewed. To further investigate this issue, we compared laboratory calibration data of both optical instruments with results from Mie theory (Figs. S4-S6). Although there are several assumptions within the Mie model that may not be totally satisfied by the laboratory generated calibration particles (e.g. spherical and homogeneous particles), the agreement with the PAX absorption and scattering measurements is fairly good for both ammonium sulfate and nigrosin. In contrast, the CAPS measurements agree well with Mie theory only for purely scattering particles, i.e. ammonium sulfate. In the case of nigrosin, the CAPS measurements agree with Mie theory





in terms of total extinction, but the measured scattering is around 43 % higher than estimated from the model. Despite several experimental efforts, we could not find the origin of the discrepancies in the CAPS scattering measurement or a way to properly correct it. Instead, we derived the CAPS scattering coefficient from the PAX absorption measurement using a thrust dependent absorption Angstrom exponent (AAE) obtained from aircraft engine measurements with a seven-

wavelength aethalometer. All the details of this calculation can be found in Sect. S1.5 in the Supplementary Information.

**2.4 Fuel specifications**

The main differences between the conventional Jet A-1 fuel and the different HEFA blends used in this work are reported in Table 1. Most fuel properties were measured following the standard ASTM (American Society for Testing and Materials) methods, including the concentration of total aromatics, naphthalene and sulfur, the smoke point and the fuel density. In

addition, the hydrogen mass concentration was determined by nuclear magnetic resonance, using a method equivalent to ASTM D7171. As expected, increasing the concentration of HEFA fuel in the blend corresponded to a reduction in the concentrations of the aromatic compounds (including naphthalenes) and sulfur. Besides, with the addition of the HEFA fuel the hydrogen mass concentration and the smoke point increased while the fuel density slightly decreased.

**2.5 Radiative forcing**

To estimate the instantaneous direct radiative effects of the PM from aircraft engine exhaust we evaluated the radiative transfer equation introduced by Chylek and Wong (1995), modified as in Bond et al. (2007), to express the wavelength dependent Simple Forcing Efficiency (SFE) in terms of the mass scattering coefficient (MSC) and the mass absorption coefficient (MAC):

$$SFE\,(\lambda) = -\frac{S_0(\lambda)}{4}T_{atm}^2(\lambda, z)(1 - F_c)\left[2\left(1 - a_s(\lambda)\right)^2\beta(\lambda)MSC(\lambda) - 4a_sMAC(\lambda)\right], \tag{1}$$

where $S_0$ is the top of the atmosphere solar irradiance, $T_{atm}$ is the atmospheric transmission, $F_c$ is the cloud fraction, $a_s$ is the surface albedo, $\beta$ is the backscatter fraction, $\lambda$ is the wavelength, and z is the height over sea level.

The MAC and MSC determined for the two measurement wavelengths (532 and 870 nm) were fitted over the entire range of the solar radiation spectrum (280-4000 nm) using the power law relationships in Eq. (2) and (3):

$$MAC\,(\lambda) = a\lambda^{-AAE} \tag{2}$$

$$MSC\,(\lambda) = b\lambda^{-SAE}, \tag{3}$$

where a and b are fitting parameters, and AAE and SAE represent the absorption and scattering Angstrom exponents, respectively. The selection of the parameters in Eq. (1) to (3) will be discussed in the results section.





## 3 Results and discussion

All the results presented in this work were corrected to the engine exit plane taking into account the dilution in the PM line and the losses in the sampling system. The $CO_2$ measurements in the Annex 16 and the PM sampling lines were used to determine the dilution factor, and a size dependent correction was developed to estimate the diffusion and thermophoretic

losses in the various sampling lines. Some additional considerations are needed regarding the representativeness of the data presented in this work. First, our results characterize the emissions at the engine exit plane from an engine operated on the ground. Thus, a correction to take into account the atmospheric conditions at flight altitude is necessary. However, as shown in Durdina et al. (2017) using data from a turbofan engine representative of modern commercially engines (Howard et al., 1996), the altitude does not significantly influence the PM size distributions. Hence we can assume the altitude correction for

the optical properties to be minor as well. In addition, most gas and particle species measured at the engine exit plane will rapidly evolve in the atmosphere and their radiative effects can largely vary from those of the direct emissions presented in this work, where the collected data corresponds to a time after emission of approximately 0.1 to 0.6 seconds (Brem et al., 2015). Additional measurements are therefore required in order to assess the plume evolution of aircraft emissions in the atmosphere. In any case, the emissions at the engine exit plane are the basis to consider the evolution of PM properties and

are therefore the baseline for diverse atmospheric modelling scenarios.

### 3.1 EC/OC analysis

An overview of the main findings from the EC/OC analysis is presented in Fig. 2. The OC concentrations were corrected to take into account the positive sampling artifact as described in the Supplementary Information (Sect. S2.1). Representative thermograms containing the EC/OC split for samples at low-, medium-, and high-thrust levels are reported in Fig. S9. In Fig.

2, panels (a)-(c) display the EC, OC and TC thrust dependencies and the changes in their concentrations associated to the use of the 32% HEFA blend in comparison to the base Jet A-1 fuel. For the Jet A-1 fuel, the concentrations of all three carbonaceous components increased with engine thrust, from a minimum of 0.1 mgTC $m^{-3}$ at taxi (~6% thrust) to a maximum of 5.6 mgTC $m^{-3}$ at take-off (~95% thrust). As the mass concentration increased with thrust, the geometric mean diameter increased from 8 nm at taxi to 40 nm at take-off (Fig. S10). The slight increase in the mass concentrations between

taxi and ground idle (~3% thrust) has been observed in previous works (Durdina et al., 2017) and is associated with a decrease in the combustion efficiency and the air/fuel ratio. The use of the 32% HEFA blend induced a clear reduction in the EC concentrations at all thrust levels, in line with previous findings (Moore et al., 2015, Brem et al., 2015; Moore et al., 2017; Schripp et al., 2018). The HEFA effect was strongest at low thrust levels, inducing a decrease in EC mass of 50-60% for thrust levels up to 30%. However, very large uncertainties were associated to the EC measurements at ground idle due to

low filter loading (down to 0.2 µg $cm^{-2}$ despite the long sampling times and the use of the small filter mask). For thrust levels of 50% and above, the HEFA effect became less significant, reaching a minimum EC decrease of 14% at take-off. A similar trend was observed for the OC and TC concentrations at high thrust levels. In contrast, the OC, and consequently also the



TC, seemed to be enhanced by the HEFA blend at ground idle. Panels (d) and (e) in Fig. 2 show the correlations of EC and OC with the BC concentration measured with the MSS ($BC_{MSS}$, reported as nvPM mass in regulatory measurements of aircraft engine emissions). EC and $BC_{MSS}$ were in excellent agreement for both fuel types, with slope very close to unity (0.96 ± 0.02) and Pearson Coefficient ($R^2$) of 0.99. Although OC also increased with BC, the correlation was weaker in this

case and small differences could be observed between the two fuels. The fraction OC/TC reported in panel (f) showed a large variability with the thrust level, with OC/TC between 0.75 and 0.90 at low thrust levels (3 - 30%), decreasing to 0.25 at 50% thrust, and down to 0.17 at take-off. Similar trends in the OC/TC ratio with engine thrust have been observed in previous works (Delhaye et al., 2017). The use of the HEFA fuel did not have any visible effect on the OC/TC, but only on the concentrations.

**3.2 Optical properties**

The main results from the measurement of the optical properties are reported in Fig. 3. For clarity, only the measurements with Jet A-1 fuel and the 32% HEFA blend are shown in this plot, while the results from the intermediate HEFA blends (5% and 10%) are included in Figs. S11 and S12. Panels (a)-(c) in Fig. 3 report the thrust dependencies of the absorption, scattering and extinction coefficients, measured at 532 nm (CAPS, green squares) and 870 nm (PAX, blue circles). At both

wavelengths, $b_{abs}$, $b_{scat}$ and $b_{ext}$ showed the same thrust dependency as EC. At most thrust levels, the three optical coefficients decreased with the HEFA blend. Panel (d) shows the correlation between $b_{abs}$ at the two measurement wavelengths and the EC mass concentration determined from the thermo-optical measurements. From the linear regressions we derived the MAC values for aircraft exhaust at 532 nm and 870 nm, which appear to be independent of the particle size distribution, thrust or HEFA concentration, and will be further discussed in the next section. $BC_{PAX}$ and $BC_{MSS}$ were strongly linearly correlated

(Fig. S13, $R^2$ =0.99, slope 0.98). The decrease in both $BC_{PAX}$ and $BC_{MSS}$ with the 32% HEFA blend is illustrated in panel (e) in Fig. 3. While a decrease in the BC mass was observed already at low concentration blends for most thrust levels (Fig. S12), the largest effect was seen with the 32% HEFA blend. Similarly to the decrease in the EC mass (Fig. 2a), the decrease due to the 32% HEFA blend was highest at low thrust levels (e.g. at 6% thrust $BC_{MSS}$ decreases by 74% and $BC_{PAX}$ by 95%), while thrust levels above 60% were characterized by a lower and rather constant decrease (around 20% decrease in both BC

measurements). The two instruments reported the same BC reduction at high thrust (high BC mass concentration), but disagreed increasingly as the BC mass concentration decreased. This was attributed to the low sensitivity and high noise of the PAX at concentrations <10 µg m$^{-3}$ relative to the MSS (error bars in panel (e)). Lastly, panel (f) reports the SSA calculated at the two measurement wavelengths as the fraction of scattering to total extinction. The high OC content of the particles at low thrust levels resulted in very high SSA, which showed a maximum at ground idle ($SSA_{CAPS,base}$ = 0.88 and

$SSA_{PAX,base}$ = 0.55). The SSA decreased sharply between 30% and 60%, likely due to decreasing OC fraction (Fig. 2f). The SSA was lowest at the combustor inlet temperatures and air/fuel ratios representative of cruise thrust (~60% thrust) where $SSA_{CAPS,base}$ = 0.29 and $SSA_{PAX,base}$ = 0.07. Above 60% thrust there was again a slight increase in the SSA, which might be related to the increasing mean particle size (as the OC content remained constant in this thrust range). While there was no





visible effect from the HEFA blend at these high thrust levels, it seems that slightly higher SSA were associated to the HEFA blend at low thrust levels.

The similarities observed in the thrust dependencies of OC/TC and SSA, as well as the high correlation between $BC_{PAX}$, $BC_{MSS}$ and EC, indicate that the OC content in these particles strongly enhanced light scattering at both measurement
wavelengths, but did not have a substantial effect on the light absorption. The increased OC content of the PM with decreasing engine power is in agreement with the observations of Vander Wal et al. (2016), and could be explained by inefficient and incomplete combustion at the lower thrust levels. The thermograms from the EC/OC analysis show a large OC volatility range for all thrust levels, with a major fraction of OC evaporating between 200 and 310 °C, especially at low thrust (Fig. S9).

**3.3 Radiative forcing**

For the calculation of the SFE defined in Eq. (1), $S_0(\lambda)$ (in W m$^{-2}$ per nm of bandwidth) was set to the synthetic reference spectrum of solar irradiance at the top of the atmosphere developed by Gueymard (2004) (Fig. S14(a)). The wavelength dependent $T_{atm}$ was evaluated for cruise conditions (z=12 km) using the Simple Model of the Atmospheric Radiative Transfer of Sunshine (SMARTS) model (version 2.9.5, Gueymard, 2001), as detailed in the Supplementary Information
(Sect. S2.4). Although previous works have proposed $F_c = 0.6$, Hassan et al. (2015) showed that Eq. (1) only performs properly when $F_c$ is set to zero, while it gives unrealistic results when clouds are present. This is because the equation assumes that aerosols below or above clouds can be neglected, which is obviously incorrect and not supported by observations. Therefore, we only considered the case of $F_c = 0$. The surface spectral albedos "Water or calm ocean", "Perennial rye grass", "Light soil", and "Fresh dry snow" incorporated in the SMARTS model from the Jet Propulsion
Laboratory Advanced Spaceborne Thermal Emission and Reflection Radiometer (ASTER) spectral reflectance database (Hook, 2018) were used to represent ground surfaces covered by sea water, grass, soil and snow, respectively. In absence of backscattering measurements, $\beta$ was fixed to 0.17 as previously determined for highly absorbing soot (SSA = 0.2) from diesel emissions (Schnaiter et al., 2003).

The wavelength dependent MAC and MSC were determined by fitting the measurements with Eq. (2) and (3), as
shown in Fig. S14(b). The MAC values at the two measurement wavelengths were determined from the linear fit between the $b_{abs}$ and the EC mass (Fig. 3(d)), which yielded $MAC_{532} = 7.5 \pm 0.3$ m$^2$ g$^{-1}$ ($R^2$=0.97) and $MAC_{870} = 5.2 \pm 0.9$ m$^2$ g$^{-1}$ ($R^2$=0.94). The latter is in good agreement with the filter based determined MAC value by Petzold and Schröder (1996) for jet engine aerosol ($MAC_{800} = 6$ m$^2$ g$^{-1}$), which using the inverse wavelength dependency of the cross section leads to $MAC_{870,calc} = 5.5$ m$^2$ g$^{-1}$. The results are also in line with the MAC value of freshly generated light absorbing carbon
proposed by Bond and Bergstrom (2007) ($MAC_{550} = 7.5 \pm 1.2$ m$^2$ g$^{-1}$), which converted to the wavelengths of interest results in $MAC_{532,calc} = 7.8 \pm 1.2$ m$^2$ g$^{-1}$ and $MAC_{870,calc} = 4.7 \pm 0.8$ m$^2$ g$^{-1}$. The MSC values were calculated using its relation with the SSA and MAC, i.e. SSA=MSC/(MSC+MAC). Using the SSA measured at 60% thrust ($SSA_{532} = 0.37 \pm 0.03$; $SSA_{870} = 0.09 \pm 0.01$) and the MAC values reported above, this calculation yielded $MSC_{532} = 4.5 \pm 0.4$ m$^2$ g$^{-1}$ and $MSC_{870} = 0.54 \pm$





0.04 $m^2 g^{-1}$. The $MSC_{532nm}$ falls within the higher end of MSCs measured for fresh biomass smoke by Levin et al. (2010). However, a more detailed comparison with literature values is hindered by the strong dependency of MSC on the particles size, morphology, and chemical composition. The AAE in Eq. (2) was set to $1.0 \pm 0.2$, as inferred for the BC emissions at cruise thrust level (60%) from the aethalometer measurements described in the Supplementary Information Sect. S1.5. This

is a widely accepted value of AAE, often used in literature for fresh black carbon particles. Lastly, SAE = $4.5 \pm 0.7$ was calculated for cruise conditions using the scattering coefficients at the two measurement wavelengths, i.e. SAE = $\ln(b_{scat,532}/b_{scat,870})/\ln(532/870)$.

The SFE spectral dependence of the PM emissions during cruise conditions is shown in Fig. 4 for the four different ground types (i.e. surface albedos) considered in this work. Positive SFE indicates a warming effect; negative SFE

corresponds to cooling. The high surface albedo of snow ($a_{snow}$ = 0.5-1.0 for $\lambda$ < 1400 nm), translated into a strong positive SFE, especially in the visible range. The highest spectral forcing over snow $SFE_{snow}$ = 16.5 $W g^{-1} nm^{-1}$ was found at the blue wavelength (450 nm). The lower surface albedos from grass and soil (on average $a_s$ = 0.2 for both surface types) induced a moderate SFE, which even turned negative (i.e. cooling) at short wavelengths ($\lambda$ < 400 nm for soil and $\lambda$ < 490 nm for grass). However, the overall dominant effect was warming, with a maximum SFE of 2.6 $W g^{-1} nm^{-1}$ found in the red visible range

(between 700-760 nm) for both surface types. In contrast, the extremely low surface albedo from sea surfaces ($a_{sea}$ = 0.004-0.04) yielded very small SFE, which was mainly negative and had a minimum of -1.7 $W g^{-1} nm^{-1}$ at 330 nm. Figure 4 also contains the integrated forcing (in $W g^{-1}$) in the spectral range 450-2000 nm (limited by the availability of albedo data) for the four surface types. The strongest warming effect of the aircraft PM was observed when the emissions occurred above highly reflective surfaces like snow. The integrated SFE in this case was in the order of 4700 $W g^{-1}$. Other land surfaces (i.e.

soil and grass) showed a moderate warming, with an integrated SFE in the range of 900 to 1600 $W g^{-1}$. The integrated effect of the emissions over dark surfaces like sea water was extremely low (~-3 $W g^{-1}$) and, contrary to the other surface types, the overall effect was cooling.

This simple model does not consider the effects from underlying clouds. However, aircraft cruising altitude (10 – 13 km above sea level) normally exceeds the typical cloud top altitude (except in tropical latitudes). Samset and Myhre (2015)

studied the differences in the modeled altitude dependence of the DRF of BC when clouds are taken into consideration, and found that globally the DRF of BC at cruise altitude (100 hPa) was doubled when using all-sky conditions (~2300 $W g^{-1}$) compared to clear-sky conditions (~1000 $W g^{-1}$). In a first approximation, we expect that cruise emissions above clouds will induce similar radiative effects to the emissions over snow covered surfaces (i.e. strong warming), but more complex models are required to accurately determine the radiative effects of aircraft PM emissions.

## 30  4 Conclusions

This work presents the EC/OC content and optical properties of PM emissions at the engine exit plane of a CFM56-7B operated at a full range of thrust levels from ground idle to take-off. Both scattering and absorption increased with thrust, as



did the PM mass. However, the light scattering was dominant at low thrust levels, while absorption prevailed otherwise. These changes were reflected in the SSA, which varied extensively over the full thrust range, and was a critical parameter for the determination of the radiative effects. The variations in the optical behavior of the particles were linked to changes in the EC/OC content. While PM at the engine exit plane is thought to mostly contain strongly absorbing EC, we found a
significant fraction of OC at low thrust levels, which explains the high scattering and SSA values.

In addition, we examined the effects of HEFA biofuel blends on the PM emissions. In line with previous studies, the 32% vol. HEFA blend significantly lowered the PM mass emissions, especially at low thrust levels, where the EC mass was reduced by 50-60%. The OC mass also decreased at most thrust levels, except at 3% thrust, where it seemed to be enhanced. However, at this thrust level we only had one sample for each fuel type, and the uncertainties attributed to these low thrust
measurements with low concentrations were large. Moreover, we could not see any effect from using the HEFA blend on the intensive optical properties (i.e. SSA, MAC or MSC), nor on the EC/OC ratio. Thus, the particles originated from the combustion of both fuel types seem to be equivalent in terms of their normalized optical properties and only their concentrations change.

The combination of optical measurements at two wavelengths enabled us to evaluate the wavelength dependency of
the optical properties, which is needed for the modeling of aerosol climate effects. For this purpose we used the SFE as an estimate of the instantaneous direct radiative forcing of the aircraft PM emissions during cruise conditions, and evaluated the differences among various surface albedos. In the absence of clouds, when the emissions occurred over dark surfaces like sea water, the forcing efficiency was very small and had a net cooling effect. In contrast, these particles had a strong warming effect when emitted above highly reflective surfaces, such as snow or ice. However, more accurate and complex climate
models that include the effect from variable underlying cloud fields are required for a complete understanding of the impact of aviation particle emissions on the Earth's radiative balance.

**Author contribution**

ME designed the study, performed the laboratory calibrations and the data analysis. BTB and FS were in charge of fuel logistics. FS coordinated the test cell availability and engine lease. ME, BTB, LD and DS performed the jet engine
measurements. AF performed the EC/OC analysis. ME wrote the manuscript with important contributions from BTB, LD and AF. BTB, LD, AF and JW revised the manuscript.

**Data availability**

The numerical data used to make the figures in this manuscript and the corresponding supplementary information is available at https://doi.org/10.5281/zenodo.1918161 (Elser et al., 2018).





**Competing interests**

The authors declare that they have no conflict of interest.

**Acknowledgements**

Funding was provided by the Swiss Federal Office of Civil Aviation (FOCA) through the project SFLV 2015-113
"EMPAIREX – EMissions of Particulate and gaseous pollutants in AIRcraft engine EXhaust". We thank Mike Weiner and
the test cell crew from SR Technics AG for operating the engine testing facility, our EMPA colleagues Regula Haag and Dr.
Daniel Rentsch for the fuel hydrogen analysis, Dr. Michael Arndt from AVL GmbH for loaning the PAX instrument, and the
group of Dr. André Prévôt from PSI for providing the Aethalometer data.

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

| Property (units) | Method | Jet A-1 | HEFA 5% | HEFA 10% | HEFA 32% |
|---|---|---|---|---|---|
| Aromatics (% v/v) | ASTM D 1319 | 18.1 | 17.1 | 16.2 | 11.3 |
| Naphthalenes (% v/v) | ASTM D 1840 | 0.79 | N.A. | N.A. | 0.53 |
| Sulfur (ppm) | ASTM D 5453 | 490 | N.A. | N.A. | 350 |
| Hydrogen mass (% m/m) | NMR | 13.8 | 13.75 | 13.81 | 14.3 |
| Smoke point (mm) | ASTM D 1322 | 22 | N.A. | N.A. | 24 |
| Density (kg m$^{-3}$) | ASTM D 4052 | 794.8 | 793.3 | 791.2 | 781.8 |

**Table 1. Fuel specifications overview (N.A.: measurement not available)**

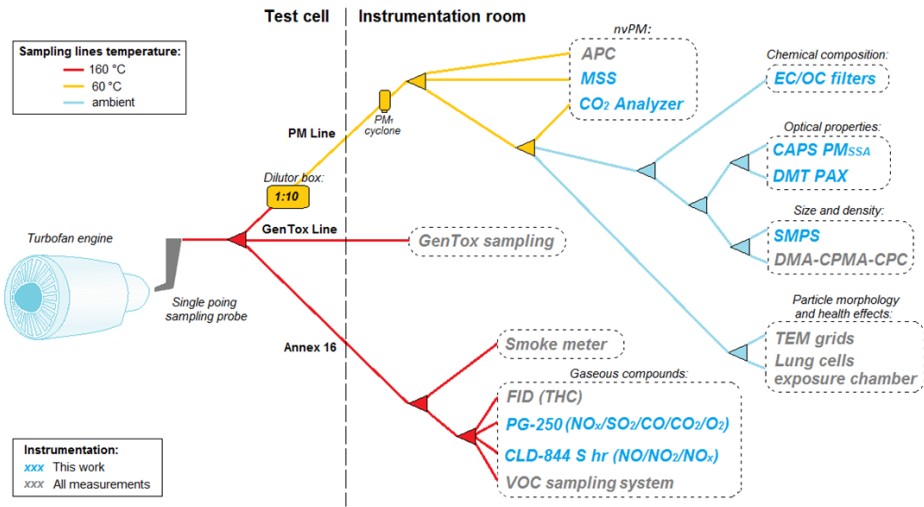

**Figure 1: Experimental setup during EMPAIREX 1. Instruments depicted in blue were used in this work, which included: a Micro Soot Sensor (MSS), a Cavity Attenuated Phase-Shift Single Scattering Monitor (CAPS PM$_{SSA}$), a Photo Acoustic Extinctiometer (DMT PAX), a Scanning Mobility Particle Sizer (SMPS), a CO$_2$ analyzer, a Portable Multi Gas Analyzer (PG-250), and a Chemiluminescence Detector (CLD-844).**



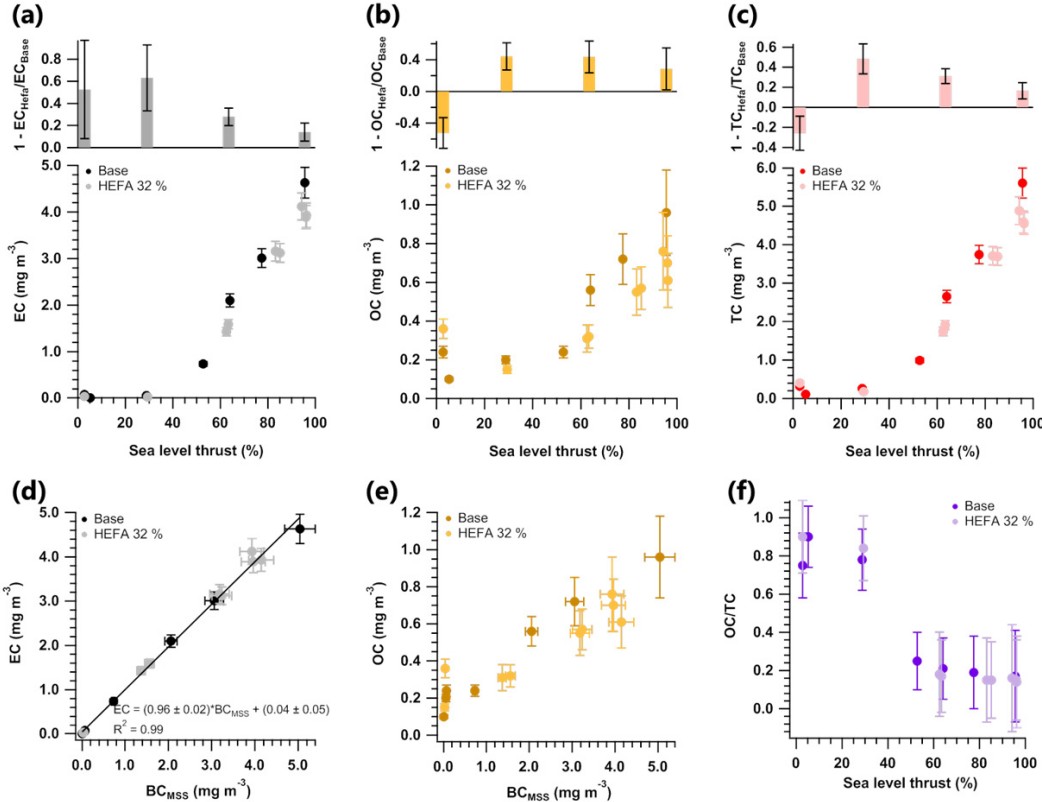

**Figure 2. Thrust dependent mass concentration and decrease with the 32% HEFA blend compared to the base Jet A-1 fuel of (a) EC, (b) OC, and (c) TC. (d): Linear fit between EC and $BC_{MSS}$ mass concentration. (e) Correlation between OC and $BC_{MSS}$ mass concentration. (f): Thrust dependent OC to TC ratio. Note: Dark colors represent measurements with base fuel (Jet A-1) and light colors represent measurements with the HEFA blend (32% vol.).**



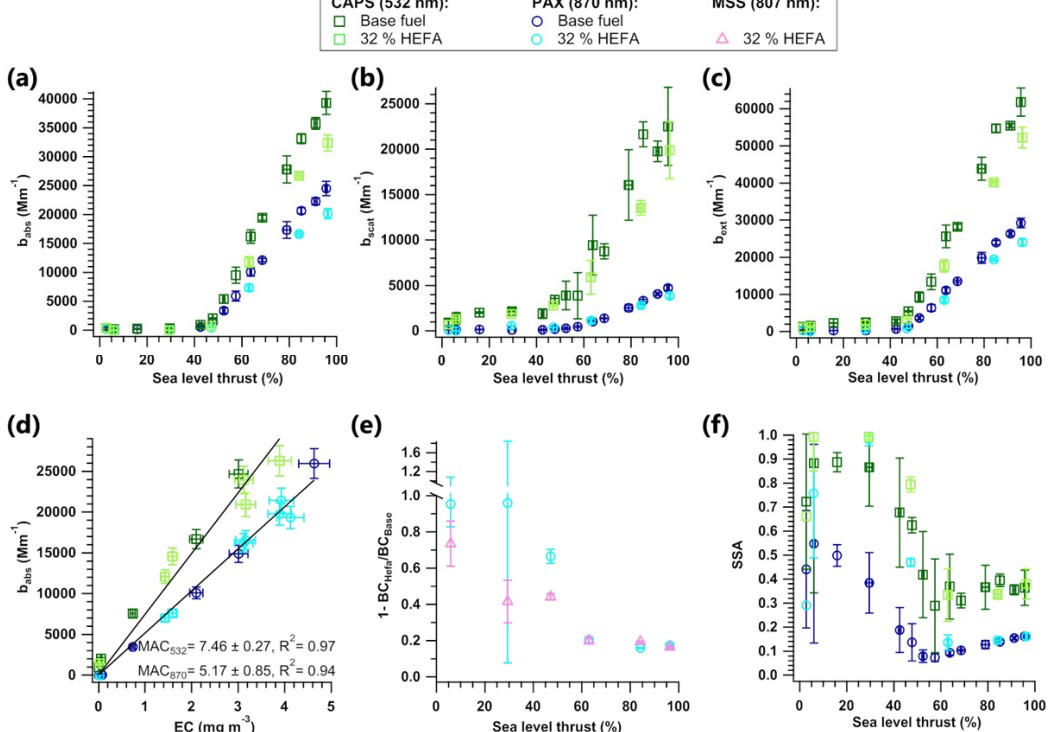

**Figure 3. Thrust dependent (a) absorption, (b) scattering and (c) extinction coefficients measured at 532 nm (green squares) and 870 nm (blue circles). (d) Correlation between the absorption coefficients and the EC mass concentration; MAC values are retrieved from the slope of the linear fits. (e) Thrust dependent decrease in $BC_{MSS}$ and $BC_{PAX}$ for the 32% HEFA blend in comparison to the base Jet A-1 fuel. (f) Thrust dependent single scattering albedo (SSA) at the two measurement wavelengths. Note: Dark colors represent measurements with base fuel (Jet A-1) and light colors represent measurements with the 32% vol. HEFA blend.**



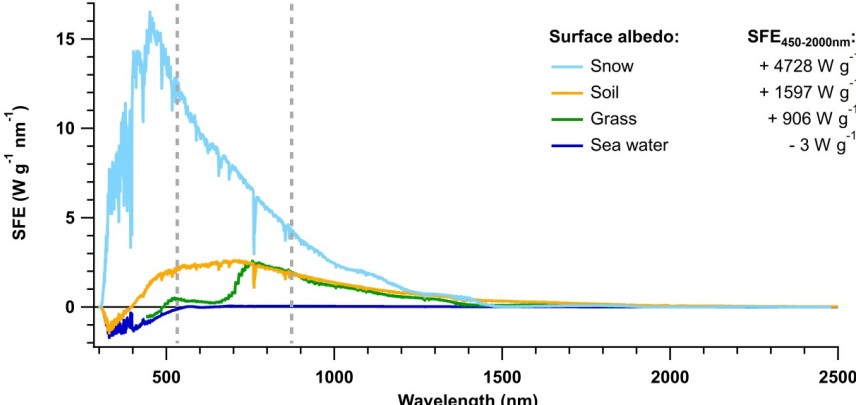

**Figure 4.** Simple forcing efficiency (SFE) spectra for aircraft engine PM over different surface types, including sea water, grass, soil and snow, and integrated spectral values for the four surface types.