# Peer review of "Chemical composition and radiative properties of nascent particulate matter emitted by an aircraft turbofan burning conventional and alternative fuels"

_Atmospheric Chemistry and Physics, 2018_

## Referee Comment (RC1) · Anonymous Referee #2 · 11 Jan 2019

The manuscript "Chemical composition, optical properties and radiative forcing efficiency of nascent particulate matter emitted by an aircraft turbofan burning conventional and alternative fuels" describes test rig measurements on a CFM56 engine using a series of different HEFA blends. Particles in the engine exhaust were characterized with filter OC/EC measurements and PAX/CAPS instruments. The results were used to estimate the radiative forcing of the particles in the atmosphere.

The manuscript covers a topic of current scientific interest and the experimental details are sound. However, some assumptions with regard to the radiative forcing are rather

bold. It is presumed that that the particles at the engine exit plane on a test rig are similar to particles behind the engines inflight. It is not clear to me if the authors considered changes to the particles in the contrails. The authors do not discuss limitations of their study but I think this is vital for the manuscript.

With regard to the impact of the HEFA blends, the authors conclude that "the particles originated from the combustion [. . .] seem to be equivalent in terms of their normalized optical properties and only their concentration change" (page 11). Huang et al. analyzed the particle morphology in the APEX III campaign. They conclude that "Such dependence upon combustion indicates that PM from alternative fuels will be different from that by JP-8. Models of PM formation in turbulent reaction environments will need to include such variations for accurate prediction. Accordingly optical properties and surface chemistry will vary too." (Huang, C.H., Bryg, V.M., Vander Wal, R.L., 2016. A survey of jet aircraft PM by TEM in APEX III. Atmospheric Environment 140, 614-622). This finding does not fit to the statement in the current manuscript. The authors are recommended to discuss this discrepancy and the uncertainty of their findings.

Overall, estimating the global impact of particles in the atmosphere based on one ground measurement of an in-service engine might be not valid enough. Nevertheless, the fuel variation experiment and its results are important for the current discussion of extended use of alternative fuels in aviation.

---

## Referee Comment (RC2) · Anonymous Referee #3 · 18 Jan 2019

This paper provides details of the chemical composition and the optical properties of the particulate matter (PM) measured at the exit plane of a CFM56-7B engine burning four blends of Jet A-1 and HEFA fuels. The paper itself is an important contribution to the literature in terms of characterizing the chemical and optical properties from an aircraft engine burning fuels with varying composition. The paper is well written but has several deficiencies that must be addressed by the authors. Chief among them is how the authors calculate the direct radiative forcing.

General Comments:

The measurements reported in this paper were performed at ground level behind the engine exit plane, but the authors have not adequately discussed the impact of different temperature and pressure regimes at cruise levels on optical properties.

Also, the impact of plume evolution at cruise conditions had not been discussed. The authors' goal of coming up with radiative effect of aircraft particulate emissions falls short in this regard.

Specific comments:

Pg 1, Ln 3: The authors switch between "aircraft particulate emissions" and "aircraft exhaust aerosol". Please be specific and consistent.

Pg 1, Ln 19: I'm not sure what "The separation of elemental carbon (EC) and organic carbon (OC)" means? Please clarify

Pg 3, Ln 25-26: Moore et al., 2017 reports on the emissions from a 50:50 blend at cruise. Please update the text to include the engine type and fuel since you have previously stated that these are important factors. Please also state that the reductions were measured at cruise levels.

Pg 3, Ln 28-29: The authors mention the engine type in the abstract. It should also be included here.

Pg 7, Ln 7-10: I don't see how you can assume that the difference in optical properties at altitude compared to ground level conditions are minor when you don't have any supporting evidence. Also, the study cited was for PM size distributions, and not nvPM size distributions.

Pg 7, Ln 13-14: The authors state "Additional measurements are therefore required in order to assess the plume evolution of aircraft emissions in the atmosphere." However, there are several studies that have reported plume evolution of aircraft engine emissions. These studies have shown a dramatic change in PM size distributions from exit plane measurements. How do the authors reconcile their approach with this published

literature data?

Pg 8, Ln 22-23: "the decrease due to the 32% HEFA blend was highest at low thrust levels". Other studies have also reported that the largest decrease was observed at low thrust levels. Can the authors comment on why this is generally the case?

Pg 8, Ln 27-33: Can the authors put the SSA results in context with measurements from other combustion sources?

Pg 9, Ln 23-24: Diesel engine emissions are significantly different from those of aircraft engines, in terms of size, density, EC/TC, etc. Is there a more appropriate source to estimate backscattering?

Pg 10, Ln 8-22: The authors are using engine exit plane measurement data for SFE estimation, and do not consider the cooling effect of sulfates. This section should either be supported with additional data or removed from the paper. Also, no information is presented on the impact of using blends of HEFA with Jet A-1 on radiative forcing.

Pg 16, Table 1: It's surprising that the Jet A-1 and 10% HEFA fuels have similar hydrogen content by different aromatic contents. Since the Jet A-1 is blended with HEFA, shouldn't the hydrogen content for 10% HEFA be higher? Likewise with the 5% blend, where the hydrogen content is lower than that of the unblended Jet A-1. Can the authors explain?

Pg 16, Figure 1: What is APC? It has not been previously defined.

Pg 17, Figure 2: Panel (f) is an interesting result (no difference in OC/TC for Jet A and 32% HEFA). Is this result unique to the fuel tested or have there been similar observations in other studies?

---

## Referee Comment (RC3) · Anonymous Referee #1 · 29 Jan 2019

The manuscript presents measurements of the organic/elemental carbon fractions and optical properties (scattering, absorption, extinction) of aircraft engine soot sampled at the exhaust plane of the engine while in a ground-based test cell. Four different fuels were examined, including both Jet A-1 and mixtures of Jet A-1 with a HEFA jet biofuel. The manuscript is relatively short and well written. There is a quite a bit of methods information in the supplementary material that should be moved to the main text to improve readability. The paper should be publishable after the following comments are addressed:

[Figure]

1) Emissions data are reported in terms of concentrations (e.g., mg m$^{-3}$ or Mm$^{-1}$), which are not particularly informative to the reader for interpreting the differences between thrust settings and fuels. The authors should normalize the results using the CO$_2$ concentration data to report the data in terms of emissions indices (e.g., mg [kg fuel]$^{-1}$ for mass emissions and m$^2$ [kg fuel]$^{-1}$ for optical coefficients). While this makes the data more useful and accessible to the readers, it won't change the intensive parameters, and therefore the conclusions of the paper.

2) It would be worthwhile to split the panels in Figures 2 and 3 into multiple figures that emphasize key relationships in the analysis. For example: 1) a new figure combining Figures 2d and 2e with Figure 3d to emphasize the MAC calculations and the contribution of EC and OC to the MSS BC mass EI. Similarly, a new figure combining Figure 2f and Figure 3f emphasizes the transition at ∼50% thrust from highly scattering, organic-carbon-dominated exhaust particles to highly absorbing, elemental-carbon-dominated exhaust particles. I would encourage the authors to think critically about how best to present these figures in order to support their discussion rather than just lump them into large, multi-panel figures with different (and not really comparable) x-axes.

3) The article is rather short as it is currently structured, so much of the additional discussion in the supplementary material could and should be moved to the main text. This is particularly true of the text in Section S1. The tables and figures in Section S1 could remain in the the supplementary material with direct referencing from the main manuscript. Section S1.4, in particular, is rather theoretical, speaks to the measurement data quality, and would be valuable to placed in the more prominent location of the main manuscript.

4) I agree with the other reviewers that the simplified radiative forcing model and calculations presented in Section 3.3 do not really add much to the paper. This is evident from the merely qualitative discussion of this section in the abstract and conclusions. One would expect *a priori* that introducing absorbing aerosols such as aircraft soot above a reflective surface would produce a warming effect and that the magnitude

of this warming effect would decrease over darker surface types. I suggest that this section be removed.

---

## Author Comment (AC1) · 13 Mar 2019

We thank all the reviewers for their valuable comments, which have greatly contributed to improving the quality of the manuscript. Our point-by-point responses (**blue**) and the proposed changes to the manuscript (**red**) are embedded below within the referees' original comments (**black**). For cases where the changes are substantial throughout the entire manuscript, we refer to the revised manuscript (with track changes) attached at the end of this response.

**Authors' Response to Anonymous Referee #1**

The manuscript presents measurements of the organic/elemental carbon fractions and optical properties (scattering, absorption, extinction) of aircraft engine soot sampled at the exhaust plane of the engine while in a ground-based test cell. Four different fuels were examined, including both Jet A-1 and mixtures of Jet A-1 with a HEFA jet biofuel. The manuscript is relatively short and well written. There is a quite a bit of methods information in the supplementary material that should be moved to the main text to improve readability. The paper should be publishable after the following comments are addressed:

**R1.1) Referee comment:**

1) Emissions data are reported in terms of concentrations (e.g., mg m$^{-3}$ or Mm$^{-1}$), which are not particularly informative to the reader for interpreting the differences between thrust settings and fuels. The authors should normalize the results using the CO2 concentration data to report the data in terms of emissions indices (e.g., mg [kg fuel]$^{-1}$ for mass emissions and m$^2$ [kg fuel]$^{-1}$ for optical coefficients). While this makes the data more useful and accessible to the readers, it won't change the intensive parameters, and therefore the conclusions of the paper.

**Author's response:**

We see the benefit of fuel normalized emissions, in particular for PM mass and EC. However, we do not believe that the normalization of optical data with fuel consumption would make the data more accessible and we think that such a normalization might confuse some readers. However, in order to make our results readily available for the modelling community, we have added in the supplementary information the thrust and fuel dependent emission index (EI) of EC (EI$_{m,EC}$), all the additional data needed to calculate EIs (i.e. $CO_2$, CO and THC concentrations) and the particles' size data (GMD and GSD). Therefore, fuel-normalized optical coefficients can easily be determined if needed and the data becomes more accessible.

**Changes in text:**

**Page 8, Line 13:** The EC, OC and TC mass concentrations are reported in Table S3. Additionally, in Table S4 we also report the mass emission index of EC (EI$_{m,EC}$, in mg kg$_{fuel}^{-1}$), together with the additional parameters required for the calculation of EIs (i.e. carbon dioxide, carbon monoxide and hydrocarbon concentrations) and the particles' size parameters (geometric mean diameter (GMD) and geometric standard deviation (GSD)).

| | Thrust (%) | $CO_2$ * (ppm) | CO ** (ppm) | THC (ppm) | $EI_{m,EC}$ (mg $kg_{fuel}^{-1}$) | GMD (nm) | GSD (nm) |
|---|---|---|---|---|---|---|---|
| **Jet A-1** | 95.6 | 45101.5 | 30.4 | 16.4 | 166.4 ± 11.8 | 42.4 ± 0.6 | 1.91 ± 0.05 |
| | 77.4 | 40793.8 | 17.2 | 9.1 | 119.8 ± 8.0 | 36.5 ± 0.3 | 2.02 ± 0.04 |
| | 64.0 | 36546.1 | 14.5 | 12.7 | 93.2 ± 6.0 | 32.1 ± 0.2 | 2.10 ± 0.03 |
| | 52.7 | 33274.3 | 12.9 | 7.7 | 36.4 ± 2.4 | 27.0 ± 0.1 | 2.12 ± 0.02 |
| | 28.8 | 28471.7 | 31.2 | 11.0 | 3.3 ± 0.6 | 17.2 ± 0.1 | 1.89 ± 0.01 |
| | 5.2* | 20726.1 | 303.6 | 71.6 | 0.9 ± 0.3 | 9.0 ± 0.1 | 1.93 ± 0.02 |
| | 2.7 | 23230.7 | 753.4 | 295.1 | 5.5 ± 0.9 | 7.9 ± 0.1 | 1.94 ± 0.02 |
| **HEFA 32%** | 96.2 | 44803.0 | 25.0 | 9.7 | 141.5 ± 9.2 | 40.1 ± 0.4 | 1.98 ± 0.04 |
| | 96.0 | 44170.7 | 22.2 | 10.2 | 141.9 ± 9.3 | 38.4 ± 0.4 | 2.02 ± 0.04 |
| | 94.2 | 44459.0 | 27.1 | 10.2 | 149.3 ± 10.7 | 39.1 ± 0.5 | 1.99 ± 0.05 |
| | 85.1 | 41531.6 | 19.0 | 9.0 | 122.6 ± 8.0 | 35.9 ± 0.3 | 2.04 ± 0.04 |
| | 83.2 | 41559.1 | 17.7 | 9.0 | 120.9 ± 7.9 | 36.6 ± 0.2 | 2.06 ± 0.03 |
| | 63.3 | 36323.7 | 13.4 | 8.0 | 70.9 ± 4.5 | 30.1 ± 0.2 | 2.13 ± 0.03 |
| | 62.5 | 36160.0 | 13.8 | 7.8 | 63.8 ± 4.1 | 29.5 ± 0.2 | 2.17 ± 0.03 |
| | 29.3 | 28237.1 | 28.3 | 7.8 | 1.7 ± 0.4 | 15.2 ± 0.1 | 1.84 ± 0.01 |
| | 2.8 | 22969.8 | 769.7 | 306.2 | 2.6 ± 1.1 | 6.6 ± 0.1 | 2.06 ± 0.02 |

* $CO_2$ measured in the PM line corrected for dilution; ** CO corrected for wet measurement conditions

**Table S4. Thrust and fuel dependent $CO_2$, CO and THC concentrations for the calculation on emission indexes, calculated emission index of EC ($EI_{m,EC}$), and particles size parameters (GMD and GSD) obtained from the fit of the size distributions.**

**R1.2) Referee comment:**

2) It would be worthwhile to split the panels in Figures 2 and 3 into multiple figures that emphasize key relationships in the analysis. For example: 1) a new figure combining Figures 2d and 2e with Figure 3d to emphasize the MAC calculations and the contribution of EC and OC to the MSS BC mass EI. Similarly, a new figure combining Figure 2f and Figure 3f emphasizes the transition at ~50% thrust from highly scattering, organic carbon-dominated exhaust particles to highly absorbing, elemental-carbon-dominated exhaust particles. I would encourage the authors to think critically about how best to present these figures in order to support their discussion rather than just lump them into large, multi-panel figures with different (and not really comparable) x-axes.

**Author's response:**

We thank the reviewer for his/her suggestion. Following his/her recommendation we have divided Figs. 2 and 3 into 5 different figures and have restructured/adapted the results and conclusions sections accordingly. The results section in the revised manuscript is divided in the following sections/figures:

3.1 Chemical composition → Fig. 2: EC, OC and TC thrust dependency + HEFA decrease
             → Fig. 3: EC and OC correlations with $BC_{MSS}$

3.2 Optical properties → Fig. 4: $b_{abs}$, $b_{scat}$ and $b_{ext}$ thrust dependency
            → Fig. 5: $b_{abs}$ vs EC (determination of MAC values)

3.3 Link between chemical composition and optical properties → Fig. 6: OC/TC and SSA thrust dependency

3.4 Radiative properties

**Changes in text:**

See revised manuscript (with track changes) at the end of this response.

**R1.3) Referee comment:**

3) The article is rather short as it is currently structured, so much of the additional discussion in the supplementary material could and should be moved to the main text. This is particularly true of the text in Section S1. The tables and figures in Section S1 could remain in the supplementary material with direct referencing from the main manuscript. Section S1.4, in particular, is rather theoretical, speaks to the measurement data quality, and would be valuable to placed in the more prominent location of the main manuscript.

**Author's response:**

Although we agree with the reviewer that some of the sections in the supplementary material contain important information (regarding for example instruments calibration and validation of the measurements), we believe that moving such detailed descriptions of experimental procedures to the main text could decrease the level of attention given to the main findings of the work. In particular:

- Section S1.1 contains details of the experimental setup that are not unique to this work and have already been reported in previous publications.
- Section S1.2 contains a detailed description of the methodology for EC/OC analysis. This is a standard methodology and therefore its description in the main text is not required.
- Section S1.3 contains the laboratory calibrations of the CAPS and PAX. Again, those are mainly standard procedures that therefore do not need to be described in detail in the main text.
- Section S1.4 contains the comparisons between laboratory calibration measurements and the estimates from Mie theory. These comparisons were performed as a data quality check, after some inconsistencies had been found in the $NO_2$ interference calibration for the CAPS instrument. Although the section is rather extensive, the only important outcome is that our CAPS scattering signal is hampered and was therefore not used in our analysis. As this is not an instrumental manuscript, we believe that including this full section to the main text would just add unnecessary complexity to the manuscript, and reporting the main findings of the comparison is enough in order to assess the data quality in the main text.
- Section S1.5 is a consequence of the results of Section S1.4 and briefly shows how we estimated the scattering coefficient at 532 nm. As for the Section S1.4, we think that adding a detailed description of this calculation in the main manuscript is not necessary.

As mentioned above, sections S1.3, S1.4 and S1.5 are of great importance for the assessment of the data quality. However, we do not think that the manuscript would benefit from a very detailed description of these sections in the main text. Instead, we only briefly introduce the measurements and main findings of each section with appropriate references to the supplementary information, so that readers that are interested in the detailed methods and procedures can easily access them.

**R1.4) Referee comment:**

4) I agree with the other reviewers that the simplified radiative forcing model and calculations presented in Section 3.3 do not really add much to the paper. This is evident from the merely qualitative discussion of this section in the abstract and conclusions. One would expect a priori that introducing absorbing aerosols such as aircraft soot above a reflective surface would produce a warming effect and that the magnitude of this warming effect would decrease over darker surface types. I suggest that this section be removed.

**Author's response:**

We thank the reviewer for his suggestion. Our goal of reporting the simple forcing efficiency was to put the determined MAC, MSC and SSA into context, by using them in a simple radiative transfer model. However, we agree with the reviewer that using this simple model without considering the plume evolution processes (which is out of the scope of this work and drastically affects the light

scattering and SSA) does not add significant value to the manuscript and the presented results could be misunderstood. Following the reviewers suggestion regarding the split of Figs. 2 and 3, we have substantially restructured the results section in the revised manuscript (see track changes below). The section "Radiative forcing" has been changed to "Radiative properties" and focuses on the determination and discussion of the MAC and MSC values. The main description of the model and its results have been moved to the supplementary information, and only the main outcomes are briefly presented at the end of this new section. In addition, at every occasion where the model results are mentioned, we explicitly note that these results are only representative of fresh emissions, while more complex models and plume evolution measurements would be required to assess the overall radiative effects of aircraft PM emissions.

**Changes in text:**

See revised manuscript (with track changes) at the end of this response.

**Authors' Response to Anonymous Referee #2**

The manuscript "Chemical composition, optical properties and radiative forcing efficiency of nascent particulate matter emitted by an aircraft turbofan burning conventional and alternative fuels" describes test rig measurements on a CFM56 engine using a series of different HEFA blends. Particles in the engine exhaust were characterized with filter OC/EC measurements and PAX/CAPS instruments. The results were used to estimate the radiative forcing of the particles in the atmosphere.

**R2.1) Referee comment:**

The manuscript covers a topic of current scientific interest and the experimental details are sound. However, some assumptions with regard to the radiative forcing are rather bold. It is presumed that that the particles at the engine exit plane on a test rig are similar to particles behind the engines inflight. It is not clear to me if the authors considered changes to the particles in the contrails. The authors do not discuss limitations of their study but I think this is vital for the manuscript.

**Author's response:**

This is a good point raised by the reviewer that needs some clarifications. The main limitations of our study are discussed at the beginning of the results section (Page 7, Line 30). While we are aware of the limitations associated to the ground measurements at the engine exit plane, the main goal of this manuscript is to report the major optical parameters of fresh aircraft PM emissions for different thrust levels and fuel types, as these are scarce in the literature and valuable for modelling studies. In the revised manuscript we have clarified how the temperature and pressure differences between ground and cruise altitude measurements can be assumed to have a negligible effect on the optical properties of the emissions. The results of a simple radiative forcing model are presented merely to put the obtained radiative properties (MAC, MSC and SSA) into context and illustrate that aircraft fresh emissions can have a significant radiative effect under certain conditions (e.g. emissions over high albedo surfaces). However, as we discuss in the manuscript, the atmospheric aging of the particles in the emission plume will affect their radiative properties, which can largely differ from those of the direct fresh emissions reported in this work. Further experimental work (plume evolution measurements) and more complex modelling studies are required to properly evaluate the plume evolution of aircraft emissions in the atmosphere and the associated radiative forcing. However, such experimental efforts are out of the scope of this manuscript. These considerations have been highlighted in the revised manuscript.

**Changes in text:**

**Page 7, Line 29:** Some additional considerations are needed regarding the representativeness of the data presented in this work. First, our results characterize the emissions at the engine exit plane from an engine operated  at the ground. Thus, a correction to take into account the atmospheric conditions (temperature and pressure) at flight altitude is in principle necessary. However, as shown in Durdina et al. (2017) using data from a turbofan engine representative of modern commercial engines (Howard et al., 1996), the altitude does not significantly influence the PM

size distributions. While ambient conditions will affect the plume evolution, the effect on the PM chemistry at the engine exit plane can be assumed to be minimal. Consequently, also the optical properties, which strongly depend on the particle size and chemical composition, would remain unvaried at the engine exit plane. Hence we  assume the altitude correction for the optical properties at the engine exit plane to be negligible.  It is important to note however, that most gas and particle species measured at the engine exit plane will evolve in the atmosphere and their radiative effects can largely vary from those of the direct emissions presented in this work, where the collected data correspond to a time after emission of approximately 0.1 to 0.6 seconds (Brem et al., 2015). Additional measurements are therefore required in order to assess the evolution of the particles' optical properties in the emission plume.  In any case, the emissions at the engine exit plane are the basis to consider the evolution of PM properties and are therefore the baseline for diverse atmospheric modelling scenarios.

**Page 12, Line 2:** However, these results need to be taken with caution, as this simple radiative model does not consider the effect of underlying clouds. Moreover, we only consider the effect of fresh PM emissions, corresponding to an approximate time after emission of less than 0.6 s, where the jet is still conserved and high temperatures prevent the condensation of volatile species. Previous studies have shown that sulfuric acid plays an important role in the formation of secondary PM in near-field aircraft plumes (Kärcher et al., 1996). Thus, plume evolution measurements of the particles' optical properties (if possible in-flight) and more complex models are needed to assess the overall radiative effects of aircraft PM emissions.

**Page 14, Line 5:** However, more accurate and complex climate models that simulate the atmospheric aging of the particles in the emission plume and take into account the effect from variable underlying cloud fields are required for a complete understanding of the impact of aviation particle emissions on the Earth's radiative balance.

**R2.2) Referee comment:**

With regard to the impact of the HEFA blends, the authors conclude that "the particles originated from the combustion seem to be equivalent in terms of their normalized optical properties and only their concentration change" (page 11). Huang et al. analyzed the particle morphology in the APEX III campaign. They conclude that "Such dependence upon combustion indicates that PM from alternative fuels will be different from that by JP-8. Models of PM formation in turbulent reaction environments will need to include such variations for accurate prediction. Accordingly optical properties and surface chemistry will vary too." (Huang, C.H., Bryg, V.M., Vander Wal, R.L., 2016. A survey of jet aircraft PM by TEM in APEX III. Atmospheric Environment 140, 614-622).This finding does not fit to the statement in the current manuscript. The authors are recommended to discuss this discrepancy and the uncertainty of their findings.

**Author's response:**

Our results show that the use of a 32 % vol. HEFA- Jet A-1 blend does not significantly influence the intensive optical properties of the emitted particles (SSA, MAC or MSC), compared to standard Jet A-1. These results are supported by the observations on the OC/TC fraction, which is directly linked to the SSA, and is also found to be independent of the type of fuel burned.

The statement in the manuscript of Huang et al. pointed out by the reviewer refers to pure alternative fuels, and does might hold for the blends of alternative fuel with Jet A-1 used in our work. The fuel effects on soot morphology were studied during the AAFEX II measurement campaign, where 4 fuel types were tested: JP-8 (similar to Jet A-1), FT (Fischer-Tropsch), HRJ (hydro-treated renewable jet), and a 50:50 blend of JP-8 and HRJ. The results, presented in Huang and Vander Wal (2013), show that due to the aromatic content in the JP-8 fuel, the soot formation starts earlier (lower temperatures) than for the synthetic fuels (FT and HRJ) that are mainly paraffinic in content (higher smoke point). The required pyrolysis reactions that delay soot formation in the case of alternative fuels, allow for greater partial premixing to occur (via turbulent mixing) compared to the JP-8. This explains the differences in the nanostructures observed form the combustion of these fuels, e.g.: at low thrust, particle cores from JP-8 lack internal structure and have high organic content while particles from FT and HRJ fuels have a clear visible core structure. In addition, for JP-8 fuel the soot nanostructure varied with engine power level (linked to the high aromatic content of the fuel and changes in the local chemical environment), while for the alternative fuels distinct and varied types of nanostructures were found irrespective of engine

operating conditions. Accordingly, the differences between JP-8 fuel and the pure alternative fuels (FT and HRJ) could cause significant differences in the optical properties of the emitted particles, but this is not necessarily the case for the blends of alternative fuel with base fuel. In fact, the 50:50 blend of JP-8 and HRJ in the work of Huang and Vander Wal (2013) exhibits a similar trend of soot nanostructure evolution to that of the JP-8 fuel. In our work we use an even lower level of alternative fuel blending (max. 32 % HEFA), which can explain why we do not observe significant differences in the optical properties of the particles emitted when burning Jet A-1 fuel and the HEFA blends. A summary of this discussion will be added in the revised manuscript (see below).

**Changes in text:**

**Page 13, Line 24:** Previous works found significant differences in the morphology of the particles emitted when burning pure alternative fuels compared to standard jet fuels, which would translate into major differences in the particles' optical properties (Huang and Vander Wal, 2013; Huang et al., 2016). However, this does not seem to be the case for blends of alternative fuels at practical ratios for widespread usage in the foreseeable future and with considerable (> 8% v/v) total aromatics content. In fact, Huang and Vander Wal (2013) found similar trends in the soot nanostructure evolution with thrust for standard jet fuel and its 50:50 blend with an alternative fuel, while the two pure biofuels tested produced distinct and varied types of nanostructures independent of the engine thrust. Huang and Vander Wal (2013) related these differences to the different degrees of turbulent mixing in the combustion chamber prior to soot formation, which is linked to the aromatic content in the fuel. Thus, soot formation from blends with up to 50% of alternative fuel is fairly similar to the one of the unblended base fuel, which results in emissions of soot particles with similar morphology, OC/TC ratios and intensive optical properties.

**R2.3) Referee comment:**

Overall, estimating the global impact of particles in the atmosphere based on one ground measurement of an in-service engine might be not valid enough. Nevertheless, the fuel variation experiment and its results are important for the current discussion of extended use of alternative fuels in aviation.

**Author's response:**

We fully agree with the reviewer that for a precise estimate of the global impact of aircraft emitted particles, additional measurements, including different engine types and plume evolution studies, are needed. Such measurements are however hindered by their high costs, e.g. for the measurements presented in this work 103 tons of fuel were burnt and 700+ working hours were invested. As discussed above, the simple radiative transfer model used in this work for engine exit plane emissions is aimed only to illustrate the potential radiative efficiency of fresh particle emissions over different surface types, but more complex models that take into account the physico-chemical evolution of the particles after emission are essential for a proper estimate of the radiative forcing of aircraft PM emissions. In the revised manuscript the section "Radiative forcing" has been changed to "Radiative properties" and focuses on the determination and discussion of the MAC and MSC values. The main description of the simple radiative model and its results have been moved to the supplementary information, and only the main outcomes are briefly presented at the end of this new section (see track changes in the manuscript).

**Authors' Response to Anonymous Referee #3**

This paper provides details of the chemical composition and the optical properties of the particulate matter (PM) measured at the exit plane of a CFM56-7B engine burning four blends of Jet A-1 and HEFA fuels. The paper itself is an important contribution to the literature in terms of characterizing the chemical and optical properties from an aircraft engine burning fuels with varying composition. The paper is well written but has several deficiencies that must be addressed by the authors. Chief among them is how the authors calculate the direct radiative forcing.

General Comments:

**R3.1) Referee comment:**

The measurements reported in this paper were performed at ground level behind the engine exit plane, but the authors have not adequately discussed the impact of different temperature and pressure regimes at cruise levels on optical properties.

**Author's response:**

This comment has been addressed above as part of the response to R2.1.

**R3.2) Referee comment:**

Also, the impact of plume evolution at cruise conditions had not been discussed. The authors' goal of coming up with radiative effect of aircraft particulate emissions falls short in this regard.

**Author's response:**

The main goal of this manuscript is to report the major optical parameters of fresh aircraft PM emissions for different thrust levels and fuel types, as these are scarce in literature and of great value for atmospheric modelling studies. The results of the simple radiative forcing model were presented merely to give an idea of the radiative effect of fresh (0.1 to 0.6 seconds after emissions) aircraft aerosol emissions occurring over different surface types (i.e. albedos). As we discuss in the manuscript, the atmospheric aging of the particles in the emission plume will affect their radiative properties (as well as morphology, size, composition…), which can largely differ from those of the direct fresh emissions reported in this work. To properly evaluate the radiative effects of the particles in the emission plume, considerable experimental work (plume evolution measurements including optical properties, if possible in-flight) and/or much more complex modelling studies are needed. Such experimental efforts were out of the scope of this manuscript. We have instead modified the manuscript to further stress that the modelling results correspond to fresh emissions and are not representative of the effects from aged aircraft emissions. We have also restructured the results section, highlighting the discussion of the retrieved radiative properties (in terms of MAC, MSC and SSA) and moving most of the modelling results to the supplementary information

**Changes in text:**

See revised manuscript (with track changes) at the end of this response.

Specific comments:

**R3.3) Referee comment:**

Pg 1, Ln 3: The authors switch between "aircraft particulate emissions" and "aircraft exhaust aerosol". Please be specific and consistent.

**Author's response:**

We thank the reviewer for pointing out this inconsistency. In the revised manuscript we have replaced "aircraft particulate emissions" with "aircraft exhaust aerosol", as the latter specifies that we refer to emissions from aircraft engines.

**Changes in text:**

**Page 1, Line 16:** Due to the lack of measurement data, the radiative forcing from aircraft exhaust aerosol particulate emissions remains uncertain.

**R3.4) Referee comment:**

Pg 1, Ln 19: I'm not sure what "The separation of elemental carbon (EC) and organic carbon (OC)" means? Please clarify

**Author's response:**

We refer to the thermal separation of the total carbon sampled on a filter into elemental and organic carbon. Although the term "separation" is widely used in this context, we will replace it in the revised manuscript to avoid confusion.

**Changes in text:**

**Page 1, Line 23:** The analysis separation of elemental carbon (EC) and organic carbon (OC) revealed a significant mass fraction of OC (up to 90%) at low thrust levels, while

**Page 4, Line 5:** The chemical characterization of the exhaust was based on the  of elemental and organic carbon (EC/OC analysis) from filter samples…

**R3.5) Referee comment:**

Pg 3, Ln 25-26: Moore et al., 2017 reports on the emissions from a 50:50 blend at cruise. Please update the text to include the engine type and fuel since you have previously stated that these are important factors. Please also state that the reductions were measured at cruise levels.

**Author's response:**

We thank the reviewer for his suggestion. We have added the missing information in the revised manuscript.

**Changes in text:**

**Page 3, Line 31:** Moreover, a recent study on in-flight cruise emissions from the NASA DC-8 turbofan engines (CFM56-2-C1) has shown that using a 50:50 blend of Jet A and a Camelina-based HEFA biofuel  reduces the  particle emissions by 50 - 70% (Moore et al., 2017).

**R3.6) Referee comment:**

Pg 3, Ln 28-29: The authors mention the engine type in the abstract. It should also be included here.

**Author's response:**

We have added this information in the revised manuscript.

**Changes in text:**

**Page 4, Line 1:** In this work we study the link between the chemical composition and the optical properties of the PM measured at the engine exit plane of a CFM56-7B turbofan  for different engine loads and HEFA fuel blends.

**R3.7) Referee comment:**

Pg 7, Ln 7-10: I don't see how you can assume that the difference in optical properties at altitude compared to ground level conditions are minor when you don't have any supporting evidence. Also, the study cited was for PM size distributions, and not nvPM size distributions.

**Author's response:**

This comment has been addressed above as part of the response to R2.1. In addition, note that our optical and EC/OC measurements are relative to the total PM emissions (only the MSS measurements are limited to the nvPM fraction). Thus, it is reasonable to consider the changes in the PM size distributions as a base to assess changes in the optical properties and the chemical composition of the emissions.

**R3.8) Referee comment:**

Pg 7, Ln 13-14: The authors state "Additional measurements are therefore required in order to assess the plume evolution of aircraft emissions in the atmosphere." However, there are several studies that have reported plume evolution of aircraft engine emissions. These studies have shown a dramatic change in PM size distributions from exit plane measurements. How do the authors reconcile their approach with this published literature data?

**Author's response:**

The aging of the particles in the emission plume will have a strong effect on the particles' size, morphology, composition, as well as on their radiative effects. Plume evolution experiments are a powerful tool to investigate such changes, but due to the very high costs of performing such measurements (especially in-flight), the amount of data reported in the literature is limited. To the best of our knowledge, this is the first work that includes a detailed characterization of the optical properties of the aircraft emissions, which is crucial to the study of the radiative effects. Although we only report the data for fresh emissions, this is the starting point for considering the aging processes. The data presented in this work should be complemented with additional measurements

of the plume evolution of the particles' optical properties and/or models to simulate the atmospheric aging of the fresh emissions and determine the radiative effects of emissions in the plume. As mentioned above, we have modified the manuscript to underline that our modelling results correspond only to fresh emissions and are not representative for aged aircraft emissions (revised manuscript with track changes at the end of this response). In addition we have also adapted the statement pointed out by the reviewer to clarify that complimentary measurements of plume evolution should include a detailed characterization of the particles' optical properties.

**Changes in text:**

**Page 8, Line 9:** Additional measurements are therefore required in order to assess the evolution of the particles' optical properties in the emission plume.

**R3.9) Referee comment:**

Pg 8, Ln 22-23: "the decrease due to the 32% HEFA blend was highest at low thrust levels". Other studies have also reported that the largest decrease was observed at low thrust levels. Can the authors comment on why this is generally the case?

**Author's response:**

As the reviewer points out this is a commonly observed behavior of rich quench lean combustors which are employed in most engines. A detailed explanation is out of the scope of this work. In simple words it can be stated that aromatic structures present in the fuel serve as the initial building blocks in the soot formation pathways at low thrust. At high thrust the high temperatures and pressures (in addition to the low air to fuel ratio) synthesize aromatic structures from fuel aliphatic species and therefore fuel chemistry is less important. A more detailed explanation on this can be found in Brem et al. 2015.

**Changes in text:**

**Page 8, Line 28:** The HEFA effect was strongest at low thrust levels, inducing a decrease in EC mass of 50-60% for thrust levels up to 30%. An explanation for this thrust dependence can be found in Brem et al. (2015).

**R3.10) Referee comment:**

Pg 8, Ln 27-33: Can the authors put the SSA results in context with measurements from other combustion sources?

**Author's response:**

Following the reviewer's suggestion we have added examples of typical SSA values reported in literature for biomass burning and traffic emissions.

**Changes in text:**

**Page 10, Line 13:** The high OC content of the particles at low thrust levels resulted in very high SSA, which showed a maximum at ground idle ($SSA_{CAPS532,base}$ = 0.88 and $SSA_{PAX870,base}$ = 0.55). Such high SSA values are common of particle emissions from biomass burning at low combustion efficiency (e.g. $SSA_{532nm}$~0.95 in wildfire emissions (Liu et al., 2014)). The SSA decreased sharply between 30% and 60% thrust, likely due to decreasing OC fraction,  reaching a minimum at the combustor inlet temperatures and air/fuel ratios representative of cruise thrust (~60% thrust), where $SSA_{CAPS532,base}$ = 0.29 and $SSA_{PAX870,base}$ = 0.07. These low SSA values are characteristic of primary on-road vehicle particle emissions (e.g. 0.22 < $SSA_{675}$ < 0.36 from tunnel measurements (Strawa et al., 2010)).

**R3.11) Referee comment:**

Pg 9, Ln 23-24: Diesel engine emissions are significantly different from those of aircraft engines, in terms of size, density, EC/TC, etc. Is there a more appropriate source to estimate backscattering?

**Author's response:**

We thank the reviewer for bringing up this important point. Measurements of the backscattering fraction (β) are quite rare in the literature and, to the best of our knowledge, they have never been reported for aviation emissions. Our choice of β was originally based on the similarities between

the SSA (or EC/TC ratios) of the particle emissions from the diesel engine ($SSA_{550nm}$=0.20 (Schnaiter et al., 2003)) and the aircraft engine at cruise conditions ($SSA_{532nm}$=0.29, this work). However, the key variable influencing the backscattering properties is the particle size, while chemical composition seems to be less important. As the particles emitted from aircraft engines are generally much smaller than those from diesel engines, our initial assumption of β might be incorrect. Therefore, we performed a rough estimate of β for aircraft fresh emissions using Mie theory. For these calculations we used the size parameters measured at cruise conditions for pure Jet A-1 fuel (GMD=29.6 and GSD=2.0 at 57.4% thrust) and the range of refractive indexes suggested by Bond and Bergstrom (2007) (i.e. m=1.75+0.63i to m=1.95+0.79i).This resulted in β = 0.27 ± 0.01, which was used to recalculate the forcing efficiencies in the revised manuscript. The effect of changing β from 0.17 to 0.27 on the estimated SFE was very small.

**Changes in text:**

**Page 11, Line 8→ Moved to SI Page 15:**

In absence of backscattering measurements, the backscattering fraction (β) was estimated with Mie theory, using the measured size parameters at cruise conditions for pure Jet A-1 fuel (GMD=29.6 and GSD=2.0 at 57.4% thrust) and the range of refractive indexes suggested by Bond and Bergstrom (2007) (i.e. m=1.75+0.63i to m=1.95+0.79i), which lead to β = 0.27 ± 0.01.

[Figure]

**Figure S15. Simple forcing efficiency (SFE) spectra for aircraft engine PM over different surface types, including sea water, grass, soil and snow, and integrated spectral values for the four surface types.**

**R3.12) Referee comment:**

Pg 10, Ln 8-22: The authors are using engine exit plane measurement data for SFE estimation, and do not consider the cooling effect of sulfates. This section should either be supported with additional data or removed from the paper. Also, no information is presented on the impact of using blends of HEFA with Jet A-1 on radiative forcing.

**Author's response:**

Our estimates of the SFE are based on the CAPS and PAX measurements, which take into account all components of the PM at the engine exit plane. As we have mentioned above, we only focus on the fresh particulate emissions at the engine exit plane, where the sulfate is only present in the gas phase. In fact, previous aerosol mass spectrometry studies performed using the same engine and sampling system have shown that inorganic nucleating species (including sulfate) are absent in the fresh conditioned exhaust of this engine (Kiliç et al., 2018; Lobo et al., 2015). Kiliç et al. also showed that the fraction of particulate sulfate (and therefore its cooling effect) becomes relevant with the plume processing in the atmosphere, but as previously mentioned, the study of the plume evolution is out of the scope of our manuscript. In the revised manuscript we have reorganized the results section in order to highlight the discussion of the radiative properties (MAC, MSC and SSA), and have moved most of the modelling results to the supplementary information. In addition, we have

clarified throughout the text that the modelling results correspond to fresh exhaust emissions and are not representative of the effects from aged aircraft emissions.

**R3.13) Referee comment:**

Pg 16, Table 1: It's surprising that the Jet A-1 and 10% HEFA fuels have similar hydrogen content by different aromatic contents. Since the Jet A-1 is blended with HEFA, shouldn't the hydrogen content for 10% HEFA be higher? Likewise with the 5% blend, where the hydrogen content is lower than that of the unblended Jet A-1. Can the authors explain?

**Author's response:**

We thank the reviewer to point this out. We have checked the values in the table and realized that the values reported corresponded to the less precise CNH method which has large uncertainties. The Table 1 has now been updated with the correct NMR values. However it has to be stated that the precision of this method is also not better than 0.1% m/m.

**Changes in text:**

| Property (units) | Method | Jet A-1 | HEFA 5% | HEFA 10% | HEFA 32% |
|---|---|---|---|---|---|
| **Aromatics (% v/v)** | ASTM D 1319 | 18.1 | 17.1 | 16.2 | 11.3 |
| **Naphthalenes (% v/v)** | ASTM D 1840 | 0.79 | N.A. | N.A. | 0.53 |
| **Sulfur (ppm)** | ASTM D 5453 | 490 | N.A. | N.A. | 350 |
| **Hydrogen mass (% m/m)** | NMR | 13.61 | 13.68 | 13.75 | 14.09 |
| **Smoke point (mm)** | ASTM D 1322 | 22 | N.A. | N.A. | 24 |
| **Density (kg m$^{-3}$)** | ASTM D 4052 | 794.8 | 793.3 | 791.2 | 781.8 |

**Table 1. Fuel specifications overview (N.A.: measurement not available)**

**R3.14) Referee comment:**

Pg 16, Figure 1: What is APC? It has not been previously defined.

**Author's response:**

APC stands for Advanced Particle Counter. In the main text we decided to include only the definitions of the instruments that were used in this work. All additional instruments are defined in the supplementary information. This information has been added in the figure caption of the revised manuscript.

**Changes in text:**

**Page 20, Line 1:** Figure 1: Experimental setup during EMPAIREX 1. Instruments depicted in blue were used in this work, which included: a Micro Soot Sensor (MSS), a Cavity Attenuated Phase-Shift Single Scattering Monitor (CAPS PM$_{SSA}$), a Photo Acoustic Extinctiometer (DMT PAX), a Scanning Mobility Particle Sizer (SMPS), a $CO_2$ analyzer, a Portable Multi Gas Analyzer (PG-250), and a Chemiluminescence Detector (CLD-844). Additional instrumentation that was not used in this work (depicted in grey) is described in the supplementary information (Sect. S1.1).

**R3.15) Referee comment:**

Pg 17, Figure 2: Panel (f) is an interesting result (no difference in OC/TC for Jet A and 32% HEFA). Is this result unique to the fuel tested or have there been similar observations in other studies?

**Author's response:**

To the best of our knowledge this is the first comparison of OC/TC ratios from the combustion of Jet A1 fuel and its blends with alternative fuels. Although we do not see any significant differences in the OC/TC trends for the two fuel types, large uncertainties are associated to these values, which makes the interpretation of the results difficult. The use of biofuels has been observed to produce a stronger reduction in EC emissions than in OC emissions, leading to an increase in the OC/TC ratio. This has been reported in numerous studies of diesel engines powered with biofuels (e.g. Popovicheva et al., 2017). In our case we only observe slightly larger OC/TC ratios for the HEFA

blend at the lowest thrust levels (3-7%), but the values for the two fuel types still compare well within the large errors bars. We also see similar trends with engine thrust and fuel type in the intensive optical properties. In fact, slightly higher SSA are observed for the HEFA blend at low thrust levels (7-50%), but also the uncertainties of the SSA (ratio of two small values) are large.

The lack of significant differences in the OC/TC ratios for the two fuels might be related to the amount of biofuel mixed in the blend. Guarieiro et al. (2017) studied the morphology of the particle emissions from a diesel engine fueled with 4%, 50%, and 100% biodiesel (B4, B50, and B100, respectively), and found almost no differences in the OC/EC ratio when using the B4 and B50 fuels, while a clear increase was observed in the OC/EC ratio for the B100 biofuel. In the same line, Huang et al. (2016) studied the morphology of the particles emitted from an aircraft engine using JP-8 fuel (similar to Jet A-1), two different biofuels (FT and HRJ) and a 50:50 blend of JP-8 and HRJ. Their results show similar trends in the thrust dependent soot nanostructure resulting from the JP-8 and the 50:50 blend with HRJ, while the two pure biofuels produced distinct and varied types of nanostructures independent of the engine thrust. They relate these differences to the different degrees of turbulent mixing in the combustion chamber prior to soot formation (which is linked to the aromatic content in the fuel). Thus, blends with up to 50% of alternative fuel might follow similar combustion processes to the unblended base fuel, which results in emissions of soot particles with similar morphology and OC/EC ratios. A part of this discussion has been added in the revised manuscript.

**Changes in text:**

**Page 13, Line 24:** Previous works found significant differences in the morphology of the particles emitted when burning pure alternative fuels compared to standard jet fuels, which would translate into major differences in the particles' optical properties (Huang and Vander Wal, 2013; Huang et al., 2016). However, this does not seem to be the case for blends of alternative fuels at practical ratios for widespread usage in the foreseeable future and with considerable (> 8% v/v) total aromatics content. In fact, Huang and Vander Wal (2013) found similar trends in the soot nanostructure evolution with thrust for standard jet fuel and its 50:50 blend with an alternative fuel, while the two pure biofuels tested produced distinct and varied types of nanostructures independent of the engine thrust. Huang and Vander Wal (2013) related these differences to the different degrees of turbulent mixing in the combustion chamber prior to soot formation, which is linked to the aromatic content in the fuel. Thus, soot formation from blends with up to 50% of alternative fuel is fairly similar to the one of the unblended base fuel, which results in emissions of soot particles with similar morphology, OC/TC ratios and intensive optical properties.

[revised manuscript text omitted]

---

## Author Response (AR2)

The authors would like to thank once again the editor and the three anonymous referees for kindly reviewing our manuscript and providing valuable suggestions that greatly improved the quality of the manuscript.

Following the editor suggestions, we have revised the use of abbreviations and the figure captions in both the article and its supplementary information. Some additional modifications have been introduced following the minor comments from Referee #3 reported below.

**Authors' Response to Anonymous Referee #3**

Our point-by-point responses (**blue**) and the proposed changes to the manuscript (**red**) are embedded below within the referees' original comments (**black**).

**Referee comment:**

Pg 3, Ln 14: "MSC (and SSA) have also a strong dependency on the particles' size" – The authors do not present any size information from this engine to support this statement.

**Author's response:**

This statement is part of the literature review on MAC and MSC values for particles emitted from combustion processes. The aim is to point out to the reader that in the size ranges characteristic of combustion particles, contrary to the MAC, the MSC has a strong dependency on particle size. Moreover, we do provide size information for this engine (in particular the geometric mean diameter and geometric standard deviation) in the supplementary information (Table S4 and Figure S10).

**Referee comment:**

Pg 3, Ln 19: "Compared to the standard Jet A-1 fuel, biofuels can have lower net $CO_2$ emissions" – Please specify that the net lower $CO_2$ emissions are on a life-cycle basis.

**Author's response:**

We thank the reviewer for pointing this out. This has been specified in the revised manuscript.

**Changes in text:**

Pg 3, Ln 19: Compared to the standard Jet A-1 fuel, biofuels can have lower net $CO_2$ emissions on a life-cycle basis.

**Referee comment:**

Pg 3, Lns 19-20: "The use of biofuel blends in airliners is regulated by the ASTM D7566" – This statement is incorrect. ASTM establishes the specification for the fuel that can be used for commercial aviation. The use of biofuel blends in not regulated by ASTM. Please update the text accordingly.

**Author's response:**

We agree with the reviewer that the ASTM is not a regulatory agency and have modified the sentence in the revised manuscript accordingly.

**Changes in text:**

The ASTM D7566 (Standard specification for aviation turbine fuel containing synthesized hydrocarbons) allows a maximum biofuel content of 50 % in Jet fuels  and sets restrictions to the blend aromatic content (minimum of 8%), lubricity, density, freezing point and viscosity (ASTM D7566-17a, 2017).

**Referee comment:**

Pg 4, Ln 15: "dilution factor $\sim$ 1:10" – is this a dilution factor (10) or dilution ratio (10:1)?

**Author's response:**

The emissions were diluted by a factor of $\sim$ 10. This has been corrected in the revised manuscript.

**Changes in text:**

Pg 4, Ln 15: The PM line was diluted with dry synthetic air (dilution factor ~ 10)…

**Referee comment:**

Pg 7, Ln 9: Change "losses in the sampling system" to "particle losses in the sampling system"

**Author's response:**

Modified in the revised manuscript.

**Changes in text:**

Pg 7, Ln 8: All the results presented in this work were corrected to the engine exit plane taking into account the dilution in the PM line and the particle losses in the sampling system.

**Referee comment:**

Pg 7, Lns 10-11: "a size dependent correction was developed to estimate the diffusion and thermophoretic losses in the various sampling lines" – Thermophoretic losses are not size dependent. Please correct the text.

**Author's response:**

We thank the reviewer for pointing this out. We have corrected this sentence in the revised manuscript.

**Changes in text:**

Pg 7, Ln 9: The $CO_2$ measurements in the Annex 16 and the PM sampling lines were used to determine the dilution factor, and a  correction was developed to estimate the  thermophoretic losses and the size dependent diffusion losses in the various sampling lines.

**Referee comment:**

Pg 8, Ln 15: "HEFA effect became less significant, reaching a minimum EC decrease of 14% at take-off" – This sentence is confusing. Please rephrase.

**Author's response:**

This sentence has been modified in the revised manuscript.

**Changes in text:**

Pg 8, Ln 14: For thrust levels of 50% and above, the decrease in EC mass with the HEFA blend  became less significant, reaching a minimum EC decrease of 14% at take-off.

**Referee comment:**

Pg 8, Lns 20-21: There appears to be an offset for the OC mass concentration in Fig 3. Can the authors explain this?

**Author's response:**

The offset in the OC-BC scatter plot indicates that at low thrust levels (up to 30%), where the BC (and EC) mass concentrations are nearly zero, there are small amounts of OC being emitted, which is in line with the high scattering/SSA obtained at these low thrust levels.

[revised manuscript text omitted]

---

## Author Response (AR3)

Dear editor,

The changes in the figure captions had been included in the new version of the manuscript but were missing in the track changes document. This has been corrected in the version below. We apologize for this mistake.

Best regards,

5  Miriam

[revised manuscript text omitted]